# Optimal Conditions for a Multimode Laser Diode with Delayed Optical Feedback in Terahertz Time-Domain Spectroscopy

**Kenji Wada [1],*, Tokihiro Kitagawa [1], Tetsuya Matsuyama [1], Koichi Okamoto [1] and Fumiyoshi Kuwashima [2]**

[1] Department of Physics and Electronics, Osaka Metropolitan University, 1-1 Gakuen-cho, Naka-ku, Sakai, Osaka 599-8531, Japan; ttookkiihhiirroo@gmail.com (T.K.); matsuyama.tetsuya@omu.ac.jp (T.M.); okamot@omu.ac.jp (K.O.)

[2] Department of Electrical and Electronic Engineering, Fukui University of Technology, 3-6-1 Gakuen, Fukui 910-8505, Japan; f7_kuwashima@outlook.jp

* Correspondence: wada.kenji@omu.ac.jp; Tel.: +81-72-254-9264

**Abstract:** Recent studies have indicated that terahertz time-domain spectroscopy (THz-TDS) can stably and efficiently acquire output spectra using an affordable and compact multimode laser diode (MMLD) with delayed optical feedback as the light source. This research focused on a numerical analysis of the optimal conditions for employing an MMLD with delayed optical feedback (a chaotic oscillating laser diode) in THz-TDS utilizing multimode rate equations. The findings revealed that the intermittent chaotic output generated by the MMLD, characterized by concurrent picosecond pulse oscillations lasting several tens of picoseconds, proved to be highly effective for THz-TDS. By appropriately setting the amounts for the injection current and optical feedback and the delay time for the optical feedback, intermittent chaotic oscillation could be attained within a considerably broad parameter range. The generation of intermittent chaotic oscillations was confirmed by observing their characteristic asymmetric spectral shapes. Moreover, both the MMLD output spectrum and the THz-TDS output spectrum exhibited consistently stable shapes at the microsecond scale, demonstrating the attractor properties inherent in an MMLD with delayed optical feedback.

**Keywords:** terahertz time-domain spectroscopy; multimode laser diode; delayed optical feedback; laser chaos; intermittent chaos; picosecond optical pulse generation; concurrent pulse oscillation; multimode rate equations

## 1. Introduction

In recent years, the field of terahertz technology has witnessed remarkable progress in both fundamental and practical investigations [1,2]. Among the crucial techniques in this field, terahertz time-domain spectroscopy (THz-TDS) stands out as an invaluable method [3,4]. It enables the precise measurement of complex permittivity at room temperature, and its roots can be traced back to the pioneering Auston switch [5]. Consequently, numerous researchers have been actively engaged in exploring this technology [6–14]. However, a major challenge associated with THz-TDS is the requirement of a large and expensive mode-locked femtosecond laser as the light source.

One of the solutions that has already been reported to address this issue is the use of a small multi-mode laser diode known as an MMLD [15–19]. An MMLD offers the advantage of being robust, providing stable output, and being cost-effective, with a range of commercially available options. It also allows for the selection of the appropriate wavelength for the photoconductive antenna being used. However, when employing an MMLD with a continuous wave (CW) operation and a relatively narrow bandwidth, the signal strength and frequency bandwidth of the THz-TDS output are constrained, and the random phases between modes further increase the noise floor, resulting in a reduced dynamic range. Another approach is to use small superluminescent diodes (SLDs), which can provide a continuous spectrum in the THz band [20,21]. However, in this case,

similar to MMLDs, their dynamic range is limited by random phase effects. Recently, a monolithic mode-locked laser diode (MLLD) was proposed, demonstrating high signal-to-noise ratio characteristics (133 dB) and high repetition rates (several tens of GHz) in THz-TDS, and it is called ultra-high repetition rate terahertz time-domain spectroscopy (UHRR-THz-TDS) [22,23]. This light source has the potential to advance THz-TDS for outdoor applications and become a standard source in this field.

Meanwhile, efforts are also being made to improve MMLD-based light sources using chaotically oscillating laser diodes (COLDs) [24–26]. COLDs, which involve the simple configuration of adding an external mirror to an MMLD resonator, have been reported to yield more stable and efficient THz-TDS outputs when compared to standalone MMLDs. However, the underlying reasons behind the effectiveness of COLDs remain unclear. The objective of this study was to numerically determine the optimal conditions for utilizing an MMLD with delayed optical feedback to achieve stable and efficient THz-TDS outputs. This investigation was conducted by employing multimode laser diode rate equations, aiming to provide valuable insights into enhancing the performance of THz-TDS systems.

This paper is organized into four sections. In Section 2, we present the multimode rate equations for the MMLD with delayed optical feedback and the THz-TDS simulation model using the MMLD with delayed optical feedback. Section 3 begins by showcasing the simulation results that classified LD oscillation states based on the coupling coefficient of the optical feedback. Following this, we discuss the characteristics of intermittent chaotic oscillation that are effective for THz-TDS. We also present the optimal parameters (pumping rate, feedback amount, and delay time) for expanding the scope of the intermittent chaotic oscillation. In the conclusion section, we offer an overview of our paper, summarizing the results obtained for the optimal conditions of an MMLD with delayed optical feedback in THz-TDS.

## 2. Methods

### 2.1. Rate Equations for a Multimode Laser Diode with Delayed Optical Feedback

We initiated the simulation process by modeling a multimode laser diode (MMLD) with delayed optical feedback, employing the below rate equations specific to MMLDs. This comprehensive model incorporated the inclusion of Langevin noise and accounted for the band filling effect, as described in previous works [27–30].

$$\dot{E}_m = \frac{1+i\alpha}{2}\left[g_m - \frac{1}{\tau_{\rm p}} + \frac{R_{\rm spn}}{S_m} + \sqrt{\frac{2R_{\rm spn}}{S_m\Delta\tau_{\rm c}}}\xi_{S_m}\right]E_m + i2\pi m\delta f E_m$$
$$+ \frac{f_{\rm c}}{2T_{\rm rt}}E_m(t-\tau_{\rm fb}) \tag{1}$$

and

$$\dot{N} = \frac{I}{eV} - \frac{N}{T_1} + \sqrt{\frac{2N}{T_1\Delta\tau_{\rm c}}}\xi_N - \sum_m\left[g_mS_m + \sqrt{\frac{2S_mR_{\rm spn}}{\Delta\tau_c}}\xi_{S_m}\right], \tag{2}$$

where $E_m$, $E_m(t-\tau_{fb})$, $S_m$, and $g_m$ are the complex optical electric field, the complex delayed optical feedback field, the photon density, and the modal gain for the $m$th mode, respectively; $N$ is the carrier density; $\alpha$ and $R_{spn}$ are the linewidth enhancement factor and the carrier density-dependent spontaneous emission coefficient, respectively; $\delta f$ is the longitudinal mode spacing; $\tau_p$ and $T_1$ are the photon lifetime and the carrier density-dependent carrier lifetime, respectively; $f_c$ is the coupling coefficient of the optical feedback fields; $T_{rt}$ and $\tau_{fb}$ are the round-trip time of the LD cavity and the delay time of the optical feedback fields, respectively; $I$, $e$, and $V$ are the injection current, the elementary electric charge, and the laser cavity volume, respectively; $\Delta\tau_c$ is the coherence time of the amplified spontaneous emission; and $\xi_{Sm}$ and $\xi_N$ are zero-mean and unit-variance Gaussian distributions, respectively, whose amplitudes are varied every $\Delta\tau_c$ as step functions [29,30].

The modal gain, $g_m$, for the $m$th mode is expressed below as a function of the differential gain coefficient, $G_{0m}$, the carrier density at transparency, $N_{0m}$, and the intrinsic gain saturation coefficient, $\varepsilon_{Nm}$.

$$g_m = \frac{G_{0m}(N - N_{0m})}{[1 + \varepsilon_{Nm}(N - N_{0m})](1 + \varepsilon_S \sum_j S_j)} \, , \tag{3}$$

where $\varepsilon_S$ is the gain compression factor and $G_{0m}$, $N_{0m}$, and $\varepsilon_{Nm}$ for the $m$th mode are represented by polynomial equations of mode number $m$ [28–30]. The condition $m = 0$ stands for the central mode, which was set to 375 THz (800 nm) by taking into account wavelength matching with a GaAs photoconductive antenna. A negative mode number indicated that the oscillation frequency for the mode of interest was lower than the central frequency, while a positive mode number indicated that it was higher. The $m$-value was varied over a range of −30 to +30, allowing for the simulation of multimode oscillation involving a total of 61 modes.

The carrier density-dependent carrier lifetime, $T_1$, and the carrier density-dependent spontaneous emission coefficient, $R_{spn}$, are expressed as follows:

$$\frac{1}{T_1} = C_1 + C_2 N + C_3 N^2 \tag{4}$$

and

$$R_{spn} = \beta C_2 N^2 \, , \tag{5}$$

where $C_1$, $C_2$, and $C_3$ are the nonradiative recombination rate, the radiative recombination coefficient, and the Auger recombination coefficient, respectively, and $\beta$ is the spontaneous emission factor.

The relation between the complex optical electric field, $E_m$, and the photon density, $S_m$, for the $m$th mode is given by:

$$S_m = \frac{n_r^2 \varepsilon_0}{2 h f_m} |E_m|^2 \, , \tag{6}$$

where $n_r$, $\varepsilon_0$, $h$, and $f_m$ represent the refractive index of the active layer of the LD, the dielectric constant for the vacuum, the Planck's constant, and the central oscillation frequency of the $m$th mode, respectively. The notations and values for these parameters are listed in Table 1.

**Table 1.** Notations and values for the parameters [30].

| Notation | Parameter | Value | Unit |
|:---:|:---:|:---:|:---:|
| $m$ | Mode number | −30~+30 | |
| $\tau_p$ | Photon lifetime | 2.0 | ps |
| $T_{rt}$ | Round-trip time of the LD cavity | 7.1 | ps |
| $\tau_{fb}$ | Delay time of the optical feedback fields | 1 | ns |
| $C_1$ | Nonradiative recombination rate | $2.0 \times 10^8$ | $s^{-1}$ |
| $C_2$ | Radiative recombination coefficient | $2.0 \times 10^{-16}$ | $m^3\ s^{-1}$ |
| $C_3$ | Auger recombination coefficient | 0 | $m^6\ s^{-1}$ |
| $f_c$ | Coupling coefficient of the optical feedback fields | 0~0.6 | |
| $\alpha$ | Linewidth enhancement factor | 3.0 | |
| $\beta$ | Spontaneous emission factor | $1.0 \times 10^{-6}$ | |
| $\varepsilon_S$ | Gain compression factor | $0.05 \times 10^{-23}$ | $m^3$ |
| $\delta f$ | Longitudinal mode spacing | 0.139 | THz |
| $\Delta \tau_c$ | Coherence time of amplified spontaneous emission | 7.1 | ps |
| $e$ | Elementary electric charge | $1.60 \times 10^{-19}$ | C |
| $L$ | Laser cavity length | 300 | µm |
| $V$ | Laser cavity volume | 180 | µm$^3$ |

**Table 1.** *Cont.*

| Notation | Parameter | Value | Unit |
|:---:|:---:|:---:|:---:|
| $n_r$ | Refractive index of the active layer | 3.6 | |
| $\varepsilon_0$ | Dielectric constant for the vacuum | $8.85 \times 10^{-12}$ | F/m |
| $H$ | Planck's constant | $6.63 \times 10^{-34}$ | Js |
| $f_m$ | Central oscillation frequency of the $m$th mode | $375 + 0.139\,m$ | THz |
| $I$ | Injection current | $r \times I_{th0}$ | mA |
| $r$ | Pumping rate | 1.5~2.5 | |
| $I_{th0}$ | Threshold current for the central mode | 24 | mA |

The delayed optical feedback term was added to the last term on the right-hand side of Equation (1) and recombined with the MMLD cavity using a coupling coefficient, $f_c$. This formulation was based on the Lang–Kobayashi equation for single-mode oscillation [31]. Since the MMLD output released approximately 90% of the optical energy from the cavity to the outside during the round-trip time, up to 81% of the optical energy was fed back into the MMLD cavity through the delayed optical feedback. As a result, the maximum value of $f_c$ was 0.9 ($= \sqrt{0.81}$). To simplify the interaction between the complex delayed optical feedback fields and the complex optical electric fields in the LD cavity, we assumed a coupling phase between them of zero, and therefore, $f_c$ was treated as a real number. This choice reflected the positioning of the external mirror at an integer multiple of the optical path length of the LD cavity.

Under the specified parameter values, we computed the temporal evolution of the variables $E_m(t)$ (61 modes) and $N(t)$ by numerically integrating the aforementioned multimode rate equations using the fourth-order Runge–Kutta method. The time interval for the numerical integration was set to approximately 0.03 ps by dividing 1 ns into $2^{15}$ (=32,768) time steps. This resulted in a frequency bandwidth of 32.768 THz that could be handled in the calculation. After calculating the initial 5 ns of the transient temporal waveforms for the variables, we recorded time-series data every 1 ns. This process was repeated 1000 times, resulting in a total time-series data collection of 1 μs. The 1 ns time series data for the 61-mode LD output, denoted as $E_m(t)$, were summed together with the random phases. Subsequently, a fast Fourier-transform was applied to obtain the complex optical spectrum data, $\tilde{E}_{sum}(\omega)$, and their absolute values were squared to record power spectrum data, $|\tilde{E}_{sum}(\omega)|^2$, every 1 ns. This procedure was repeated 1000 times, resulting in a dataset comprising 1000 power spectra.

### 2.2. Simulation Model for THz Time-Domain Spectroscopy

Figure 1 illustrates the typical configuration for a THz-TDS system. The system comprised an MMLD with delayed optical feedback, serving as the input light source. The input light was divided equally into pump light and probe light using a half mirror. The pump light was focused on a gap between the metallic electrodes of a photoconductive antenna, generating photo-carriers, $n(t)$, which is described by the following equation:

$$n(t) \propto \left| \sum_{m=-30}^{30} E_m(t) \exp(i\varphi_m) \right|^2, \tag{7}$$

where $\varphi_m$ represents an arbitrary phase for the $m$th mode. The photo-carriers induced a transient current, $J(t)$, when a bias voltage was applied to the gap. Consequently, electromagnetic waves were emitted into free space, with their amplitudes being proportional to the time derivative of the transient current, $dJ(t)/dt$. In cases where an antenna exhibits wideband characteristics due to the ultrafast relaxation times of the photo-carriers, the THz wave, $E_{THz}(t)$, can be represented as follows:

$$E_{\text{THz}}(t) \propto \frac{dJ(t)}{dt} \propto \frac{dn(t)}{dt} \, . \tag{8}$$

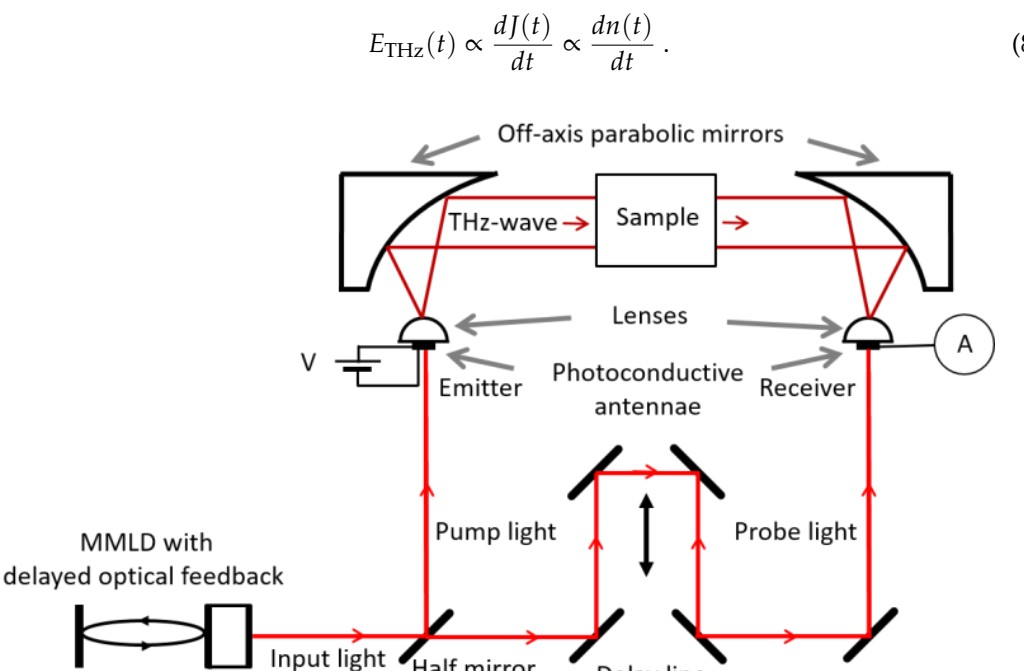

**Figure 1.** Configuration of the THz-TDS system using an MMLD with delayed optical feedback.

The frequencies of the electromagnetic waves were in the range of sub-THz to a few THz, corresponding to the optical beats of the MMLD output.

The THz waves were collimated using an off-axis parabolic mirror, and subsequently, they passed through a sample. Another off-axis parabolic mirror was employed to focus the transmitted THz waves onto a second photoconductive antenna, similar to the first one. The second antenna was also irradiated by the probe light to generate photo-carriers, $n(t)$, within its gap. The photo-carriers flowed outside the photoconductive antenna as a transient current, which was proportional to the potential difference across the gap determined by the electric field of the irradiated THz waves. By varying the delay time, $\tau$, for the probe light, a cross-correlation trace, $CC(\tau)$, between the THz and the photo-carrier waveforms could be obtained. The $CC(\tau)$ can be represented as follows:

$$CC(\tau) = \int \frac{dn(t)}{dt} n(t - \tau) dt \, . \tag{9}$$

The THz-TDS output of the no sample case, $\widetilde{CC}(\omega)$, was obtained by Fourier-transforming $CC(\tau)$ as follows:

$$\widetilde{CC}(\omega) = \int CC(\tau) \exp(-i\omega\tau) d\tau = i\omega |\tilde{n}(\omega)|^2 \, , \tag{10}$$

which was proportional to the power spectrum of the photo-carriers [30]. The THz-TDS output data were recorded at 1 ns intervals by applying a fast Fourier-transform to the 1 ns time-series data of the 61-mode LD output, which were obtained by substituting them into Equation (7) and then substituting the result into Equation (10).

### 3. Results and Discussion

*3.1. Classification of the LD Oscillation State by the Coupling Coefficient of the Optical Feedback*

Figure 2 shows the simulated results of the temporal waveforms for 1 μs of the MMLD outputs (a1–d1), along with the corresponding power spectra (a2–d2) and THz-TDS outputs (a3–d3) for the following different values of $f_c$: 0 (a1–a3), 0.1 (b1–b3), 0.4 (c1–c3), and 0.6 (d1–d3), where the pumping rate $r$ and the delay time of the optical feedback $\tau_{\text{fb}}$

were set to 1.5 and 1 ns, respectively. The power spectra of the MMLD outputs (a2–d2) and the THz-TDS outputs (a3–d3) were obtained by averaging 1000 spectral data points, each acquired at 1 ns intervals. When $f_c$ was set to zero (no optical feedback), the MMLD exhibited steady-state oscillation (a1), albeit with a relatively significant amount of noise resulting from the Langevin noise. The side-mode suppression ratio attained 26.4 dB (a2), indicating that the MMLD operated in single-mode oscillation at the negative first mode. As a result, the generation of optical beats between the multimode, based on a 139 GHz longitudinal mode spacing, was effectively suppressed, leading to a low THz-TDS output (a3). When $f_c$ was set to 0.1, the temporal waveform exhibited chaotic behavior (b1), and the spectrum demonstrated multimode oscillation, primarily operating in four modes ranging from the negative fourth to the negative first modes (b2). This behavior arose from the periodic optical feedback, which acted as a seed light for each mode, preventing the concentration of gain in a particular mode. The presence of optical feedback disrupted the MMLD's ability to settle into steady-state oscillation, resulting in a continuous transient state. Consequently, the THz-TDS output spectrum (b3) showed an increase in the 139, 278, and 417 GHz spectral components, corresponding to one, two, and three times the longitudinal mode spacing of the MMLD, respectively. Subsequently, when $f_c$ was raised to 0.4, the temporal waveform exhibited intermittent chaotic oscillation, characterized by intense peaks at the following specific time points: 6, 116, 134, 189, 284, 420, 508, 579, 648, 736, 875, and 959 ns (c1). Concurrently, the multimode spectrum of the MMLD output broadened asymmetrically (c2), leading to a corresponding broadening of the THz-TDS output spectrum to approximately 2 THz (c3). Moreover, the widening of each linewidth observed in both the LD output spectrum (c2) and the corresponding THz-TDS output spectrum (c3) indicated that the intermittent chaotic oscillation was accompanied by the presence of short optical pulses. However, when $f_c$ was further increased to 0.6, the MMLD returned to steady-state oscillation (d1) with two (the negative fifth and negative sixth) modes (d2). The resulting THz-TDS output spectrum displayed only a 139 GHz spectral component (d3). This result could be attributed to the delicate equilibrium between the optical feedback and the optical gain, with the presence of relatively strong optical feedback leading to the phenomenon of optical injection locking in the MMLD. By performing area integration on the spectra presented in Figure 2(a2–d2), we observed moderate increases in the MMLD outputs by factors of 1.06 ($f_c = 0.1$), 1.46 ($f_c = 0.4$), and 1.91 ($f_c = 0.6$) compared to the case without optical feedback (a2). These increases were attributed to the presence of optical feedback. In contrast, upon analyzing the corresponding THz-TDS outputs from the area integration of the spectra in Figure 2(a3–d3), a significant surge was estimated. The THz-TDS outputs experienced increases by factors of 22 ($f_c = 0.1$), 676 ($f_c = 0.4$), and 466 ($f_c = 0.6$) due to the effect of optical feedback. These findings demonstrated that an appropriate amount of optical feedback was necessary to achieve broadband and high-intensity THz-TDS outputs using an MMLD with delayed optical feedback. In this numerical calculation, while environmental and equipment noise were not included, a noise floor was generated based on Langevin noise. As a result, the signal-to-noise ratios of the THz-TDS outputs shown in (a3–d3) were estimated to be 5 to 9 dB lower than those obtained without Langevin noise.

The preliminary experiments involving optical feedback using various commercially available Fabry–Perot LDs confirmed the same trend as illustrated in Figure 2, albeit with variations in the extent of broadening in the multimode spectrum.

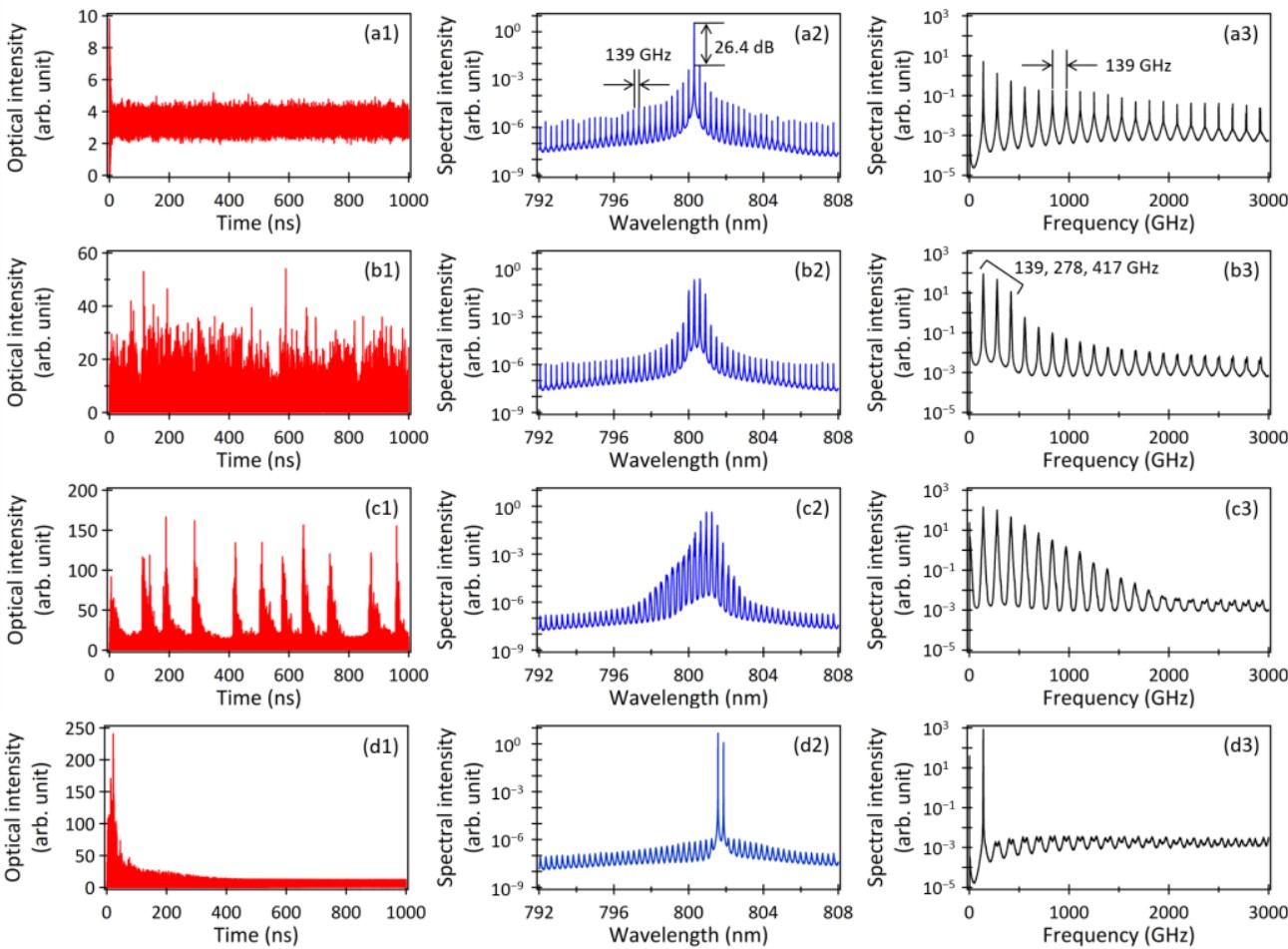

**Figure 2.** Temporal waveforms of the MMLD outputs (**a1–d1**), along with the corresponding power spectra (**a2–d2**) and THz-TDS outputs (**a3–d3**) for the different values of $f_c$ (0 (**a1–a3**), 0.1 (**b1–b3**), 0.4 (**c1–c3**), and 0.6 (**d1–d3**)). The parameters $r$ and $\tau_{fb}$ were set to 1.5 and 1 ns, respectively.

### 3.2. Characteristics of the Intermittent Chaotic Oscillations

Next, we conducted a detailed numerical investigation of the intermittent chaotic oscillation under the condition of $f_c = 0.4$. Figure 3a depicts the temporal waveform of the carrier density when $f_c$ was set to 0.4, revealing significant accumulations of carrier densities at the following specific time points: 0, 109, 130, 179, 276, 414, 502, 576, 644, 728, 866, and 956 ns. It is worth noting that the intense peak intensities observed in the MMLD output shown in Figure 2(c1) occurred between 3 and 10 ns after the carrier density accumulation timings, indicating that the MMLD operated similar to a Q-switched laser [32] under the given optical feedback condition. Figure 3b,c illustrates the temporal variations in the MMLD output energy (b) and the THz-TDS output energy (c), respectively, integrated every 1 ns. The distinct energy drops observed in Figure 3b precisely corresponded to the significant carrier density accumulations shown in Figure 3a. These rapid energy drops shown in Figure 3b could be observed when the temporal waveform, depicted in Figure 2(c1), was measured with a slightly slower optical detector, commonly referred to as "low-frequency fluctuation (LFF)" in the field of laser chaos [33]. In contrast, the distinct energy increases observed in Figure 3c aligned precisely with the strong peak intensities of the MMLD output shown in Figure 2(c1).

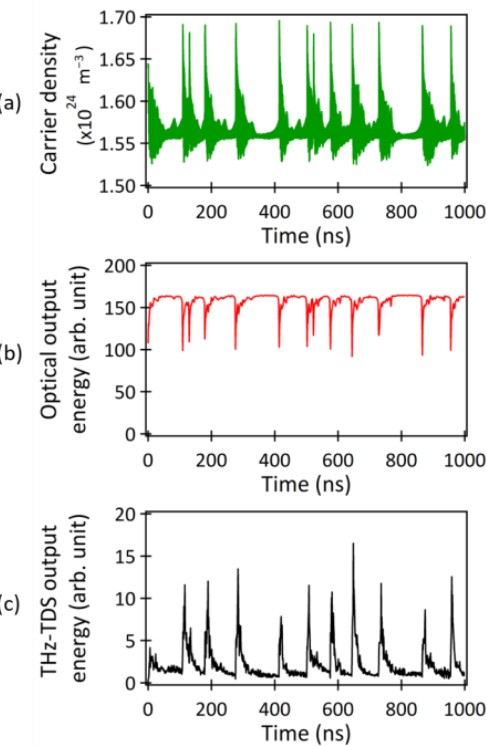

**Figure 3.** (**a**) Temporal waveform of the carrier density when $f_c$ was set to 0.4. The temporal variations in the (**b**) MMLD output energy and (**c**) THz-TDS output energy integrated every 1 ns.

In Figure 4a, a magnified view of a section from Figure 3c is presented. The points (b), (c), and (d) within the figure correspond to the THz-TDS output energies during the following specific time periods: (b) 644–645 ns, (c) 648–649 ns, and (d) 690–691 ns, respectively. Figure 4b–d displays the corresponding temporal waveforms of the MMLD output for the respective time periods. During the time periods (b) and (c), characterized by the occurrence of the LFF phenomenon, optical pulses with durations in the range of several tens of picoseconds were generated. Conversely, as shown in Figure 4d, there were periods when we observed chaotic oscillations with low peak intensities, equivalent to those of the standalone MMLD.

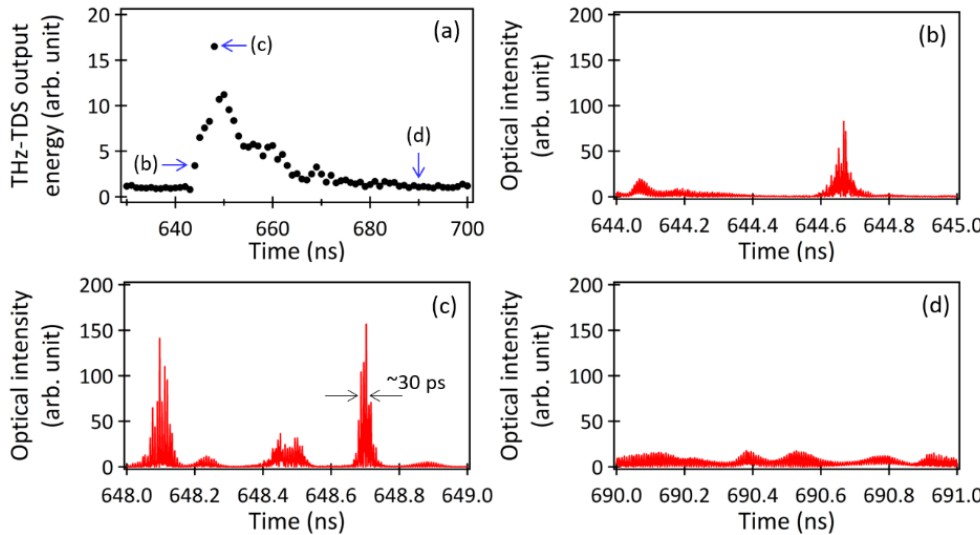

**Figure 4.** (**a**) A magnified view of a section from Figure 3c. The temporal waveforms of the MMLD outputs are shown during the time periods (**b**) 644–645 ns, (**c**) 648–649 ns, and (**d**) 690–691 ns, respectively.

Notably, the THz-TDS output energy was maximized when the shortest pulse, approximately 30 ps in duration, was generated (Figure 4c). Therefore, the efficient generation of optical short pulses was crucial for achieving effective THz-TDS utilizing the MMLD with delayed optical feedback. This was because the generation of the THz waves relied on the time derivative of the temporal waveform of the photo-carriers, as described in Equation (8).

In Figure 5a, the power spectrum of the MMLD output in Figure 4b is presented. Despite the short 1 ns time period, many oscillation mode components were generated simultaneously. The corresponding temporal waveforms of eight modes (the negative second to the fifth) are displayed in Figure 5b, revealing the nearly simultaneous oscillations of approximately 50 ps optical pulses with different pulse peaks. Thus, the optical modulation of the MMLD by the optical feedback resulted in the generation and concurrent oscillations of picosecond optical pulses, thereby enhancing the occurrence of the optical beats in THz-TDS.

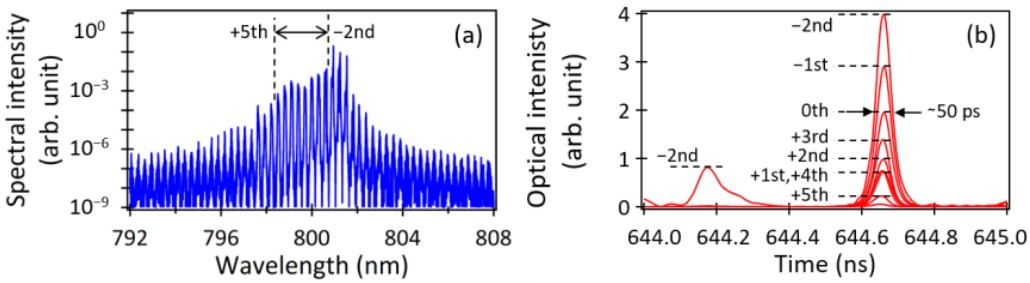

**Figure 5.** (**a**) Power spectrum of the MMLD output corresponding to the temporal waveform in Figure 4b. (**b**) Temporal waveforms of the MMLD output for the respective (the negative second to the fifth) modes.

Figure 6a,b presents the power spectra of the MMLD outputs in Figure 4c,d, respectively. Despite both spectra exhibiting symmetric shapes of the multimode oscillation due to the influence of optical feedback [34,35], there was a notable distinction in the spectral width depending on the presence or absence of high-peak picosecond optical pulses during each time period. The spectral width of the power spectrum shown in Figure 6b was comparable to that shown in Figure 2(b2), which was calculated under the condition of $f_c = 0.1$. In contrast, the spectrum in Figure 6a was approximately three times wider than that shown in Figure 6b. Furthermore, the spectral peak shown in Figure 6a had shifted toward shorter wavelengths by approximately 1 nm in comparison to that shown in Figure 6b. This shift reflected the band-filling effect resulting from the abrupt accumulations of carrier densities just prior to the generation of the picosecond optical pulses, as seen in Figure 3a. As shown in Figure 4b, the temporal waveform exhibited a mixture of picosecond pulses and chaotic oscillations with relatively low intensities, with both elements present in nearly equal proportions. As a result, the corresponding power spectrum shown in Figure 5a clearly resembled the composite shape of the power spectra shown in Figure 6a,b, with significant asymmetry in its spectral profile. The power spectrum averaged over 1 μs, as shown in Figure 2(c2), and it also exhibited a similar asymmetric shape, which was evidence that the MMLD output exhibited intermittent chaotic oscillations.

Figure 6c presents an experimental result of the output spectrum of the MMLD (Sharp, Sakai, Japan, DL-7140-213X, 780 nm) with delayed optical feedback, which was utilized as an efficient light source for generating and detecting THz waves in reference [26]. In the experiment, the injection current of the MMLD was set to 80 mA against the threshold current of 40 mA ($r = 2.0$), and an external cavity length of approximately 30 cm ($\tau_{fb} = 2$ ns) was used. The observed asymmetric spectral shape depicted in Figure 6c was consistent with the calculation shown in Figure 2(c2), indicating that the MMLD with delayed optical feedback used in the experiment exhibited intermittent chaotic oscillations, including picosecond optical pulse generation. In this sense, the COLD (chaotic oscillating

laser diode) used in the THz-TDS experiment could be specifically referred to as ICOLD (intermittent chaotic oscillating laser diode).

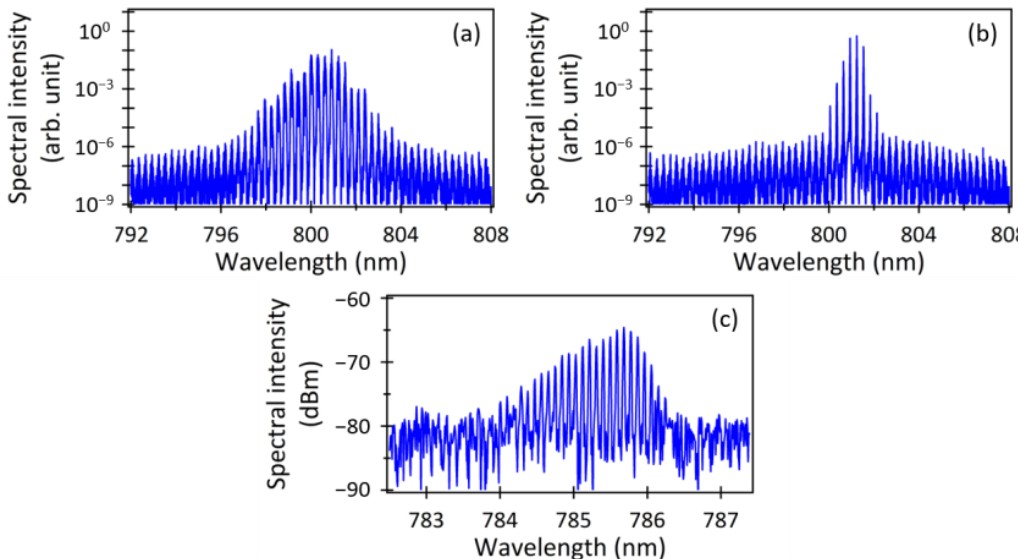

**Figure 6.** (**a**,**b**) Power spectra calculated using the temporal waveforms shown in Figure 4c,d, respectively. (**c**) Experimentally observed power spectrum of the output from the MMLD with delayed optical feedback [30].

*3.3. Time Convergence of the Output Spectral Shapes*

Figure 7a shows the temporal variation in the intensity difference, $D_{\mathrm{LD}}(t)$, between the adjacent MMLD output spectra, which were averaged every 1 ns using the rate equations mentioned above. This intensity difference was calculated using the following equation:

$$D_{\mathrm{LD}}(t_1) = \sqrt{\sum_{\lambda=792\ \mathrm{nm}}^{808\ \mathrm{nm}} \left| \frac{1}{t_1} \left| \tilde{E}_{t1}(\lambda) \right|^2 - \frac{1}{t_0} \left| \tilde{E}_{t0}(\lambda) \right|^2 \right|^2 }, \tag{11}$$

where $\frac{1}{t_0}\left|\tilde{E}_{t0}(\lambda)\right|^2$ and $\frac{1}{t_1}\left|\tilde{E}_{t1}(\lambda)\right|^2$ represent the spectral intensities of the power spectra of the MMLD outputs averaged within the time points $t_0$ and $t_1$, respectively. The time difference between $t_0$ and $t_1$ was set to 1 ns, and this calculation continued until $t_1$ reached 1 μs. The intensity difference demonstrated a convergence trend, approximately reaching a reciprocal time. Figure 7b–d depict the power spectra of the MMLD outputs averaged over 10 ns, 100 ns, and 500 ns, respectively. Taking into account the convergence of the intensity difference displayed in Figure 7a, the power spectrum shape in Figure 7d (averaged over 500 ns) closely resembled that shown in Figure 2(c2) (averaged over 1000 ns). Hence, despite the chaotic changes in the temporal waveform of the MMLD output with delayed optical feedback from moment to moment, the averaged power spectrum consistently converged to a constant shape, reflecting the attractor characteristics of the MMLD system with delayed optical feedback. These findings were consistent with the previously reported experimental result that the averaged power spectrum was stably observed with a constant shape [26]. The results presented in Figure 7 demonstrate that the power spectrum of the MMLD with delayed optical feedback stabilized to a constant shape through a time average in the order of microseconds.

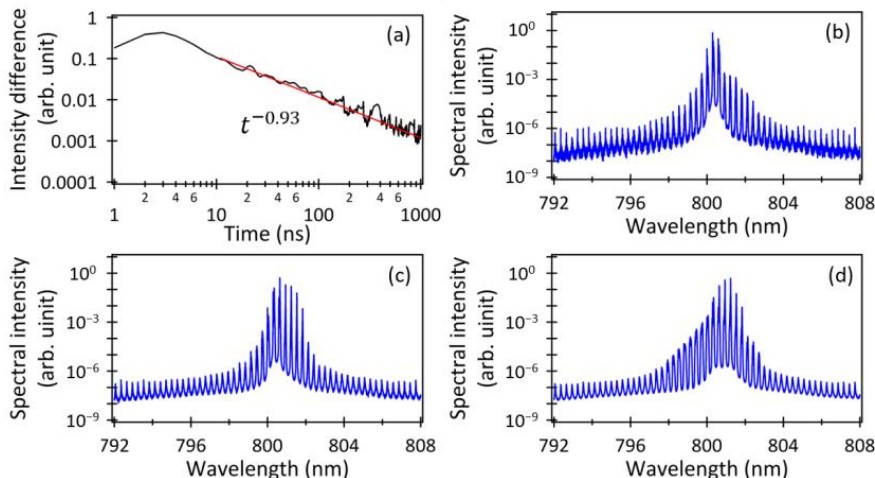

**Figure 7.** (**a**) Temporal variation in the intensity difference between the adjacent averaged MMLD output spectra. Examples of the MMLD output spectra averaged over (**b**) 10 ns, (**c**) 100 ns, and (**d**) 500 ns, respectively.

Figure 8a shows the temporal variation in the intensity difference, $D_{THz}(t)$, between the adjacent THz-TDS output spectra averaged every 1 ns. This intensity difference was calculated using the following equation:

$$D_{THz}(t_1) = \sqrt{\sum_{f=0}^{3000\,\text{GHz}} \left| \frac{1}{t_1}\left|\widetilde{CC}_{t1}(f)\right| - \frac{1}{t_0}\left|\widetilde{CC}_{t0}(f)\right| \right|^2}, \quad (12)$$

where $\frac{1}{t_0}\left|\widetilde{CC}_{t0}(f)\right|$ and $\frac{1}{t_1}\left|\widetilde{CC}_{t1}(f)\right|$ represent the spectral intensities of the power spectra of the THz-TDS outputs averaged within the time points $t_0$ and $t_1$, respectively. Similar to Figure 7, the time difference between $t_0$ and $t_1$ was set to 1 ns, and this calculation continued until $t_1$ reached 1 μs. Additionally, Figure 8b–d presents the power spectra of the THz-TDS outputs averaged over 10 ns, 100 ns, and 500 ns, respectively. The observed fluctuations shown in Figure 8a could be attributed to the time derivative of the chaotic temporal waveform of the photo-carriers during the THz wave generation described in Equation (8).

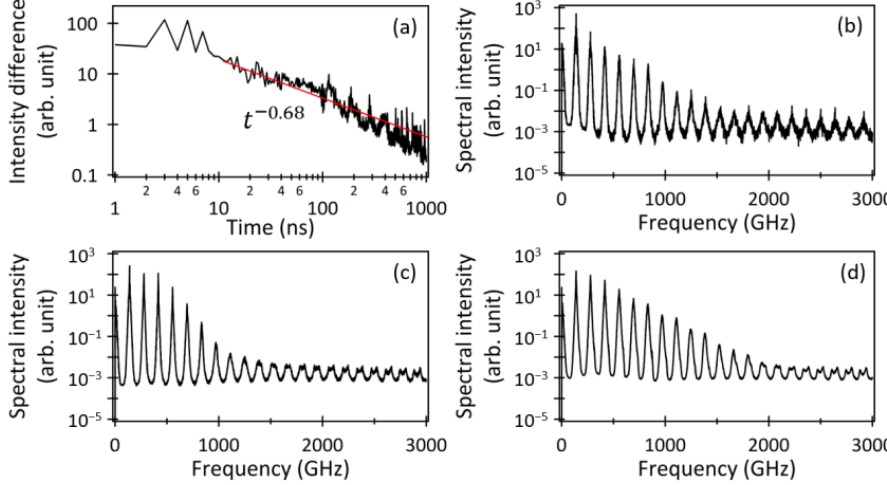

**Figure 8.** (**a**) Temporal variation in the intensity difference between the adjacent averaged THz-TDS output spectra. Examples of the THz-TDS output spectra averaged over (**b**) 10 ns, (**c**) 100 ns, and (**d**) 500 ns, respectively.

Though the fluctuations introduced a certain amount of delay, they still adhered to the overall convergence trend observed in the MMLD output spectrum illustrated in Figure 7.

### 3.4. Expansion of the Intermittent Chaotic Oscillation Region

Figure 9a illustrates the temporal waveform peaks as functions of $f_c$ for the time period 900–1000 ns, with $r = 1.5$ and $\tau_{fb} = 1$ ns (the same conditions shown in Figure 2). Notably, significant peak fluctuations with intensities exceeding 100 (arb. unit), indicative of intermittent chaos, were observed only when $f_c$ was set to 0.27, 0.32, and 0.40. These values corresponded to situations where even slight variations in $f_c$ caused sharp decreases in the THz-TDS outputs. However, previous experiments reported that the THz-TDS outputs exhibited only minor fluctuations in response to slight variations in $f_c$ [26]. To reconcile these calculation results with the experimental results, the controllable parameter values (the pumping rate, $r$, and the delay time of the optical feedback, $\tau_{fb}$) were systematically varied, and the corresponding bifurcation diagrams were plotted, as shown in Figure 9(b1–c2). Specifically, Figure 9(b1,b2) demonstrates that under higher $r$ values, the region of intermittent chaos expanded towards the higher $f_c$ side. This expansion was attributed to the increase in the relaxation oscillation frequency of the MMLD resulting from an increase in the pumping rate, $r$. The relaxation oscillation frequency of the MMLD could be expressed as Equation (13), introducing mode dependence in the form of the relaxation oscillation frequency of the single-mode LD [36].

$$f_{Rm} = \frac{1}{2\pi} \sqrt{\frac{1}{\tau_p T_1} \frac{I - I_{thm}}{I_{thm} - I_{0m}}}\,, \tag{13}$$

where $f_{Rm}$, $I_{thm}$, and $I_{0m}$ represent the relaxation oscillation frequency, the threshold current, and the current level corresponding to the carrier density at transparency, respectively, for the $m$th mode. Thus, at high excitation, the relaxation oscillation frequency was higher and fast intermittent chaotic oscillations with picosecond pulse oscillations were more likely to occur. Figure 9(c1) demonstrates that when $\tau_{fb}$ was set to a short time (e.g., 0.5 ns), the region of intermittent chaos completely disappeared. This phenomenon could be attributed to the high frequency of the optical modulation caused by the optical feedback ($1/\tau_{fb}$~2 GHz), which approached the relaxation oscillation frequency of the MMLD driven at $r = 1.5$ (~4 GHz). In contrast, as shown in Figure 9(c2), when $\tau_{fb}$ was set to 2 ns, the frequency of the optical modulation ($1/\tau_{fb}$~0.5 GHz) became significantly lower than the relaxation oscillation frequency, resulting in an expansion of the intermittent chaos region. Thus, increasing the injection current or delay time expanded the intermittent chaotic area of the MMLD with delayed optical feedback.

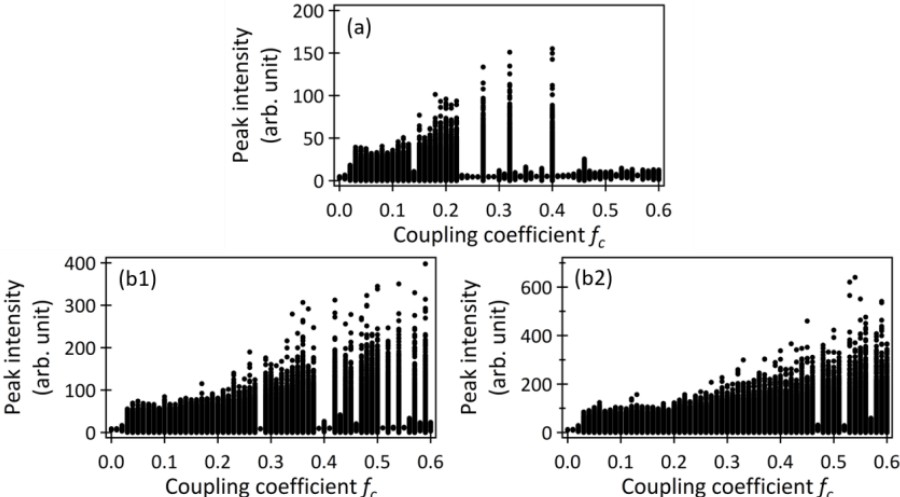

**Figure 9.** *Cont.*

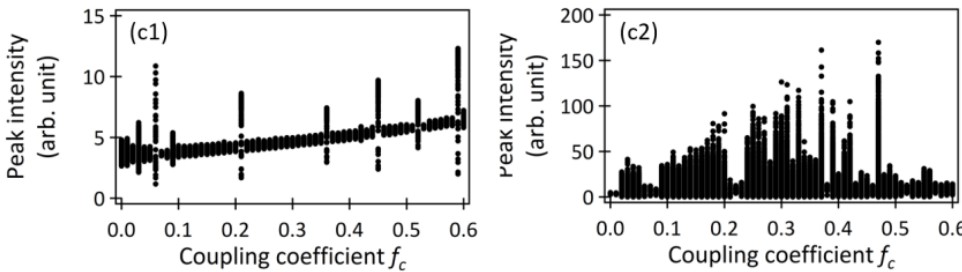

**Figure 9.** Peak plots as functions of $f_c$ when the values of $(r, \tau_{fb})$ were set to (**a**) (1.5, 1 ns), (**b1**) (2.0, 1 ns), (**b2**) (2.5, 1 ns), (**c1**) (1.5, 0.5 ns), and (**c2**) (1.5, 2 ns), respectively.

## 4. Conclusions

In the THz-TDS, we employed a miniature MMLD with delayed optical feedback as the light source. Through a numerical analysis using multimode rate equations, we identified the optimal conditions for achieving more efficient and wider bandwidth THz-TDS outputs with an MMLD with delayed optical feedback in comparison to a standalone MMLD. Remarkably, we discovered that the most favorable MMLD output for THz-TDS involved intermittent chaotic oscillation, encompassing concurrent picosecond pulse oscillations. The generation of intermittent chaotic oscillation could be confirmed by observing its characteristic asymmetric spectral shape. Furthermore, under the parameters of the MMLD with delayed optical feedback, we showed that the averaged power spectrum converged to a constant shape within the microsecond range. Correspondingly, the shape of the THz-TDS output spectrum also exhibited a similar convergence. Notably, by significantly increasing both the controllable pumping rate and the delay time of the optical feedback, we found an expansion of the parameter space in which intermittent chaotic oscillation occurred. These findings hold significant importance for the application of THz-TDS utilizing a compact and cost-effective MMLD with delayed optical feedback as the light source.

**Author Contributions:** Conceptualization, K.W. and F.K.; methodology, K.W. and F.K.; software, K.W. and T.M.; validation, K.W., K.O. and F.K.; formal analysis, K.W. and F.K.; investigation, K.W., T.K. and F.K.; resources, K.W.; data curation, K.W. and T.K.; writing—original draft preparation, K.W.; writing—review and editing, K.W., T.M., K.O. and F.K.; visualization, K.W. and T.M.; supervision, K.O. and F.K.; project administration, K.W.; funding acquisition, K.W. All authors have read and agreed to the published version of the manuscript.

**Funding:** This research received no external funding.

**Institutional Review Board Statement:** Not applicable.

**Informed Consent Statement:** Not applicable.

**Data Availability Statement:** The data presented in this paper may be obtained by submitting a request to the corresponding author.

**Acknowledgments:** The authors thank Ken Umeno of Kyoto University, Kei Inoue of Sanyo-Onoda City University, and Yuki Kawakami of NIT, Fukui College, for their fruitful discussions about chaotic oscillations from laser diodes.

**Conflicts of Interest:** The authors declare no conflict of interest.

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
