# Peer review of "Optimal Conditions for a Multimode Laser Diode with Delayed Optical Feedback in Terahertz Time-Domain Spectroscopy"

_2813-446X, doi:10.3390/spectroscj1030012_

Round 1
Reviewer 1 Report
Comments and Suggestions for Authors
This article is well-written and well-developed. It numerically investigated the optical conditions for employing an NMLD with delayed optical feedback in THz-TDS. This article only needs a minor revision before publication.
1. As an article, the introduction section is too short to reflect the advancements and disadvantages of the selected research field (terahertz technology). For example, the advantages of MMLD and the limitations of COLD can be demonstrated more clearly.
2. The meaning of the notation for Equations 1-6 should be given immediately after the equation. Similarly, Table 1 should be mentioned earlier.
3. What is the meaning of 139GHz in Figure a2 and Figure a3?
4. Line 127: For readers’ convenience, mark the 139, 278, and 417 GHz spectral components in Figure b3.
5. Line 178: What is the meaning of ‘during many other time periods’?
6. The detailed optimal conditions for MMLD with delayed optical feedback should be demonstrated more clearly in the abstract and conclusion, which is the main contribution of this article.
Author Response
To Reviewer 1
Thank you for taking the time to review our paper, especially considering your busy schedule. We greatly appreciate your valuable comments. The paper has been revised in accordance with your suggestions, and you can find the revisions highlighted in yellow within the updated version. We have also provided responses to each of your comments in the attached PDF file. Please review them at your convenience.
Kenji Wada,
Osaka Metropolitan University, Japan

Reviewer 2 Report
Comments and Suggestions for Authors
The authors provide a thorough and well-written mathematical analysis of the operating principle of a multimode laser diode (MMLD) with delayed optical feedback and its dependence on the system parameters. They present comprehensive numerical simulation results and analyze the consequences of these results for the application of an MMLD as the light source of a THz-TDS system.
However, a couple of points should be improved before publication of the manuscript.
General suggestions:
1) The authors provide an explanation of why an MMLD with delayed optical feedback providing intermittent chaotic oscillation provides better performance in a THz-TDS system than an MMLD without optical feedback.
An alternative (and comparable) light source for a THz-TDS is a monolithic mode-locked laser diode (MLLD). This approach (typically called ultra-high repetition rate terahertz time-domain spectroscopy, UHRR-THz-TDS) has recently proven great potential.
May I suggest that the authors include a comparison to the case of a mode-locked laser diode in their manuscript. While a suitable mention in the introduction may be sufficient, the inclusion of the mode-locked case in the numerical simulation would (if possible) greatly add to the value of the manuscript.
2) The authors present simulation results, in particular spectral amplitudes, in arbitrary units. I have no issue with that. However, may I suggest that the authors comment on the effect on the signal-to-noise power ratio in the resulting terahertz spectrum?.
3) How can these results be transferred to other types of laser chips?
Line-specific suggestions:
Line 37: The authors write "... which offers promising advantages". Please name in the manuscript what these advantages are.
Line 38: It is explained later in the manuscript what the difference between an MMLD and a COLD is. Please include a (short) preliminary explanation already in the introduction for the authors to better follow along.
Line 44: Please end the introduction with an overview of the structure of the paper.
Lines 51-76: The authors provide one "block" of equations followed by one "block" of text explaining these equations. May I suggest that the authors re-arrange these lines, so that every equation is immediately followed by its explanation. That would greatly enhance readability.
Table 1: I assume that the units in rows 3 and 4 are "ps" instead of "Ps"? Moreover, I assume that the unit of the laser cavity length is "mm" instead of "Mm"?
Line 112: (important) Please explain briefly how the numerical simulation was performed, i.e., how did the authors get from the mathematical equations to the numerical simulation results? Which methods and tools were used?
Figure 3: May I suggest that the authors arrange the three plots vertically, so that the time axes of the three plots are aligned.
Line 317: The authors state that the data is contained within the article. For the data provided in Table 1, I can agree with that statement. For the simulation results, I would argue that the plots merely present the results. True data availability should be provided through other means.
Author Response
To Reviewer 2
Thank you for taking the time to review our paper, especially considering your busy schedule. We greatly appreciate your valuable comments. The paper has been revised in accordance with your suggestions, and you can find the revisions highlighted in yellow within the updated version. We have also provided responses to each of your comments in the attached PDF file. Please review them at your convenience.
Kenji Wada,
Osaka Metropolitan University, Japan

Reviewer 3 Report
Comments and Suggestions for Authors
This study elucidated the optimal parameters for achieving high sensitivity and wide-band THz-TDS output using a multimode laser diode (MMLD) with delayed optical feedback. The investigation was conducted through numerical analysis employing multimode rate equations. These findings have significant implications for the application of THz-TDS that employs a compact and cost-effective MMLD with delayed optical feedback as the light source and should be published following reviewer responses.
Comments that require addressing include the following:
In line 83, we read that “Figure 1 illustrates a typical numerical model of a THz-TDS system”. What does “numerical” mean, please? I think that Figure 1 might not adequately embody the concept of a "numerical model".
.
In Figure 1, The input light is divided equally into pump light and probe light using a half mirror. In this system, it is inquired whether optical feedback is established of both the pump light and the probe light, or if it is confined to just one of them.
In lines 199-200, we read that “Despite both spectra exhibiting symmetric shapes of multimode oscillation due to the influence of optical feedback”. I suggest adding references to illustrate the effect of optical feedback, for example, Cheng Tong et al, Acta Phys. Sin. 2022, 71(6): 064205.
In lines 295-297, we read that “we determined the optimal conditions for the MMLD with delayed optical feedback to achieve highly sensitive and wide-band THz-TDS output.”. The authors are invited to expound upon their understanding of "sensitive" and how this concept is manifested in the context of this study.
In lines 278-280, we read that “This expansion is attributed to the increase in the relaxation oscillation frequency of the MMLD resulting from an increase in r, which in turn increases the likelihood of intermittent chaotic oscillation.” What does this mean? How the "likelihood" referred to can be assessed.
I’m not sure that I have read anywhere a claim for setting the amounts of injection current reported here. I do feel the additional discussion of injection current is required.

Author Response
To Reviewer 3
Thank you for taking the time to review our paper, especially considering your busy schedule. We greatly appreciate your valuable comments. The paper has been revised in accordance with your suggestions, and you can find the revisions highlighted in yellow within the updated version. We have also provided responses to each of your comments in the attached PDF file. Please review them at your convenience.
Kenji Wada,
Osaka Metropolitan University, Japan
